# Regulation of stem/progenitor cell maintenance by BMP5 in prostate homeostasis and cancer initiation

Mathieu Tremblay, Sophie Viala[†], Maxwell ER Shafer[†], Adda-Lee Graham-Paquin, Chloe Liu, Maxime Bouchard*

Goodman Cancer Research Centre and Department of Biochemistry, McGill University, Montreal, Canada

**Abstract** Tissue homeostasis relies on the fine regulation between stem and progenitor cell maintenance and lineage commitment. In the adult prostate, stem cells have been identified in both basal and luminal cell compartments. However, basal stem/progenitor cell homeostasis is still poorly understood. We show that basal stem/progenitor cell maintenance is regulated by a balance between BMP5 self-renewal signal and GATA3 dampening activity. Deleting *Gata3* enhances adult prostate stem/progenitor cells self-renewal capacity in both organoid and allograft assays. This phenotype results from a local increase in BMP5 activity in basal cells as shown by the impaired self-renewal capacity of *Bmp5*-deficient stem/progenitor cells. Strikingly, *Bmp5* gene inactivation or BMP signaling inhibition with a small molecule inhibitor are also sufficient to delay prostate and skin cancer initiation of *Pten*-deficient mice. Together, these results establish BMP5 as a key regulator of basal prostate stem cell homeostasis and identifies a potential therapeutic approach against *Pten*-deficient cancers.

**\*For correspondence:**
maxime.bouchard@mcgill.ca

[†]These authors contributed equally to this work

**Competing interests:** The authors declare that no competing interests exist.

## Introduction

Maintaining homeostasis in adult tissues requires a fine balance between stem cell maintenance and differentiation (*Morrison and Kimble, 2006*). An amplification of the stem/progenitor cell pool at the expense of cell differentiation or, conversely, the depletion of the stem/progenitor cell pool by premature differentiation are both expected to be detrimental to tissue homeostasis (*Signer and Morrison, 2013*; *Tomasetti and Vogelstein, 2015*). The dynamic process of stem/progenitor cell maintenance is regulated by signals from the microenvironment (niche) and by intrinsic regulatory mechanisms (*Dumont et al., 2015*). Similarly, cancer initiation and progression are thought to rely in large part on the self-renewal potential of cancer stem cells (*Batlle and Clevers, 2017*). In many systems, including the prostate, the molecular and cellular mechanisms regulating stem/progenitor cell homeostasis are still poorly understood.

The prostate consists of a multitude of branched epithelial ducts funneling prostatic fluids to the upper urethra. Prostatic ducts are composed of an inner layer of secretory luminal cells surrounded by a layer of basal cells interspersed with rare neuroendocrine cells. Tissue homeostasis and regeneration of the prostate have been shown to rely on endogenous stem cell populations. Allograft assays and lineage-tracing experiments revealed the presence of stem cells in both the basal and luminal compartments (*Choi et al., 2012*; *Goldstein et al., 2010*; *Wang et al., 2013*). During homeostasis and regeneration, adult stem cells are largely unipotent (*Choi et al., 2012*; *Liu et al., 2011*; *Wang et al., 2013*). However, basal stem cells act as facultative stem cells by activating basal to luminal lineage conversion under various stress conditions (*Kwon et al., 2014*; *Toivanen et al., 2016*). In contrast, basal stem cells are constitutively multipotent in the developing prostate, where they populate the luminal cell layer through asymmetric cell divisions (*Ousset et al., 2012*;

*Shafer et al., 2017*; *Wang et al., 2014a*). In prostate cancer, basal and luminal stem cells can both act as tumor-initiating cells and are thought to contribute to tumor recurrence (*Choi et al., 2012*; *Goldstein et al., 2010*; *Wang et al., 2009*; *Wang et al., 2013*; *Zhao et al., 2018*).

In recent years, GATA transcription factors have emerged as important regulators of prostate development and cancer (*Nguyen et al., 2013*; *Shafer et al., 2017*; *Xiao et al., 2016*). During post-natal development, GATA3 is expressed in both basal and luminal cells and plays an important role in lineage specification and tissue organization (*Shafer et al., 2017*). In cancer, nuclear loss of GATA3 is associated with the progression to castration-resistant prostate cancer (CRPC) and poor prognosis (*Nguyen et al., 2013*). Accordingly, *Pten*-deficient mouse model of prostate cancer exhibits a progressive loss of *Gata3* expression (*Nguyen et al., 2013*). In this model, prostate cancer could be accelerated by an acute loss of *Gata3*, or significantly delayed by GATA3 maintenance through transgenic expression (*Nguyen et al., 2013*). Despite this growing body of evidence of the importance of GATA3 in the prostate, its role in adult prostate stem/progenitor cell homeostasis is currently unknown.

Among the established regulators of stem cell homeostasis is the BMP signaling pathway (*Oshimori and Fuchs, 2012*). BMPs are part of the TGFβ family of signaling molecules, a group comprised of key regulators of cell differentiation, apoptosis, epithelial-mesenchymal transition and stem cell homeostasis (*David and Massagué, 2018*). BMPs are typically expressed in the stem cell niche and promote either stem cell quiescence or self-renewal depending on context (*Genander et al., 2014*; *Haramis et al., 2004*; *He et al., 2004*; *Tadokoro et al., 2016*; *Tian and Jiang, 2014*). In prostate cancer, BMP6 signaling appears to promote cancer progression (*Darby et al., 2008*; *Lu et al., 2017*), while BMP7 induces quiescence in metastatic cancer cells (*Kobayashi et al., 2011*). Yet, possible regulation of adult prostate stem cells by BMPs has received little attention so far.

Here, we explore the mechanisms of adult prostate stem/progenitor cell homeostasis. Combining mouse genetics, organoid cultures and RNA-seq analysis, we identify a crucial role for GATA3 in the control of basal stem/progenitor cell maintenance and identify BMP5 as a key effector of this activity. We further demonstrate that targeting the BMP pathway with a small molecule inhibitor significantly slows down cancer progression in the prostate as well as in the skin.

## Results

### Gata3 deficiency increases the long-term maintenance of prostate stem/progenitor cells

*Gata3* has previously been reported to play a role in prostate development (*Shafer et al., 2017*) and in cancer progression (*Nguyen et al., 2013*). Whether *Gata3* also regulates adult prostate stem cell homeostasis remains unknown. To explore this possibility, we first examined the expression pattern of *Gata3* in adult prostate lineages using the surface markers Lin(CD31,TER119,CD45); SCA1; CD49f; EpCAM; TROP2-, to separate basal, luminal and stromal cells (*Figure 1—figure supplement 1A,B and C*). Taking advantage of a *Gata3*$^{GFP}$ knock-in reporter mouse strain, we found *Gata3*-driven GFP expression in both the basal and luminal populations (*Figure 1—figure supplement 1B*). This finding was confirmed by qRT-PCR analysis (*Figure 1—figure supplement 1C*) and by immunofluorescence staining of GATA3 in basal and luminal cells (*Figure 1—figure supplement 1D*).

We next sought to determine whether GATA3 plays a role in prostate stem/progenitor cell homeostasis. For this, we purified basal cells from wild type, *Pbsn-Cre Gata3*$^{f/f}$ or *Pbsn-Cre Rosa26*$^{G3/G3}$ adult prostates and performed a short-term organoid propagation assay where cultures were passaged after 7 days in order to specifically look at their propagation potential. *Pbsn-Cre* mice express the Cre recombinase in both basal and luminal cells of the prostate (*Figure 1—figure supplement 1E*; *Wu et al., 2011*). In this assay, short-term organoids grown from wild type basal cells could be passaged for three to five generations, as the organoids progressively lose their propagation potential (*Figure 1A–B*). Interestingly, cells obtained from *Pbsn-Cre Rosa26*$^{G3/G3}$ mice, which express higher levels of GATA3 upon Cre-mediated deletion of a stop cassette (*Nguyen et al., 2013*), had a reduced organoid propagation potential (*Figure 1A* and *Figure 1—figure supplement 2A*). In striking contrast, cells derived from *Gata3*-deficient prostates (*Pbsn-Cre Gata3*$^{f/f}$) (*Grote et al., 2006*) showed an increased organoid-forming potential over several passages (*Figure 1A* and *Figure 1—figure supplement 2A-B*). Organoid size, proliferation and apoptosis

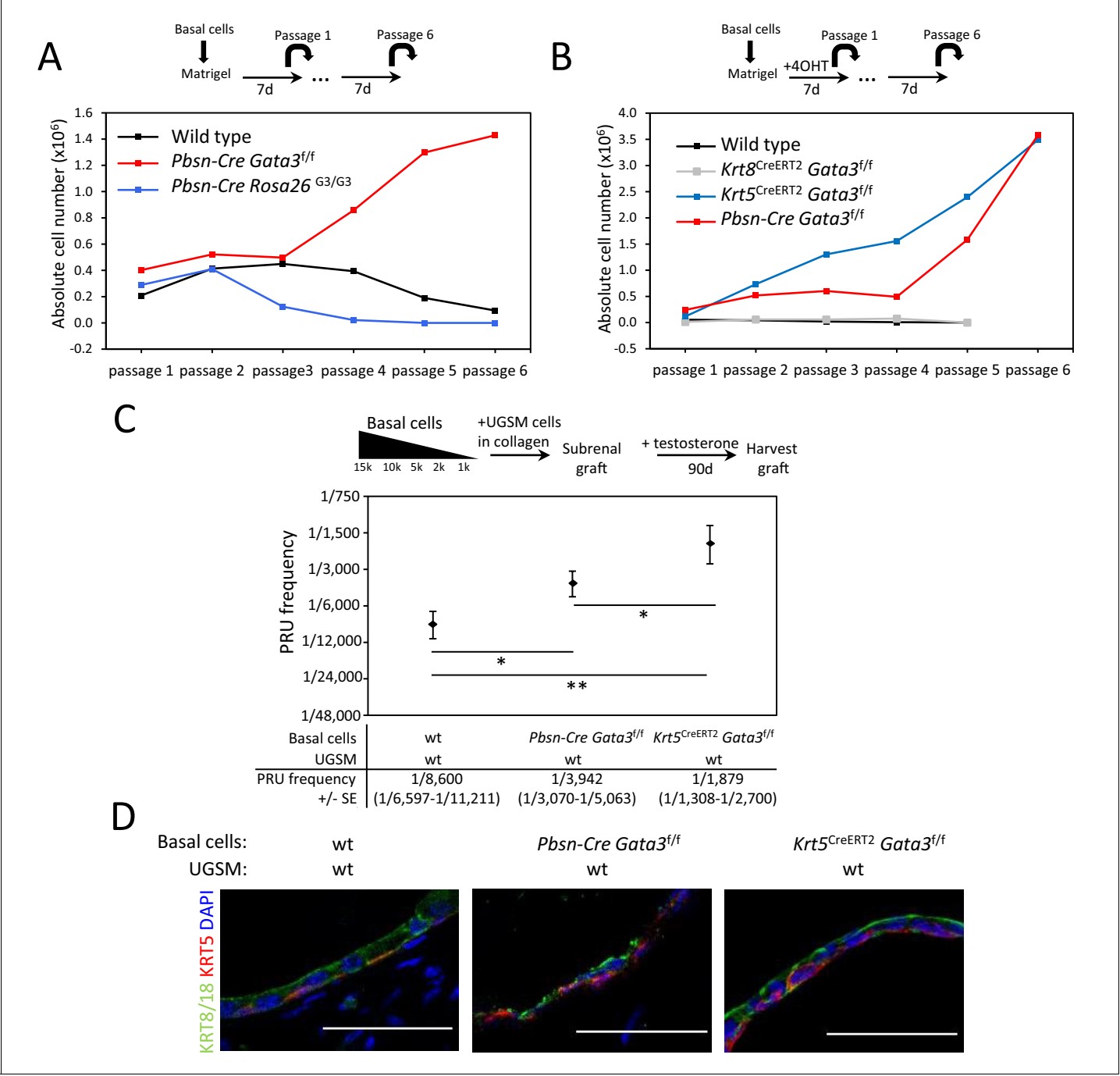

**Figure 1.** *Gata3* loss leads to an expansion of prostate basal stem/progenitor cells numbers. (**A**) Effect of *Gata3* loss and overexpression on in vitro basal stem/progenitor maintenance potential. Organoid-forming potential was assessed by plating equal numbers ($10^5$ cells) of sorted basal prostate cells from the indicated genotypes in Matrigel and passaged every 7 days. Shown is the absolute number of cells obtained after each passage for the indicated genotypes. Data are representative of two independent experiments from a pool of prostate cells from a minimum of three mice. (**B**) Specific deletion of *Gata3* in KRT5+ basal cells affect the organoid-forming potential upon passage. Organoid-forming potential was assessed as in (**A**). Cre activity was induced in vitro by treatment with hydroxy-tamoxifen for the first passage. (**C**) *Gata3* loss increase regenerative capacity in vivo. Different numbers of sorted basal cells from wild type (*Pbsn-Cre*, *Pbsn-Cre Gata3*f/f and *Krt5*CreERT2*Gata3*f/f) prostate were mixed with UGSM and transplanted under the kidney capsule of immunodeficient mice. All mice contain *Rosa26*LstopLTdTomato/+ allele and Cre activity was induced in vivo by tamoxifen injection in adult mice 4 weeks prior to organoid propagation potential assessment. Prostate reconstituting units (PRU) frequency of total basal cells was calculated based on growth of TdTomato+ grafts using the Limiting Dilution Analysis software L-Calc (StemCell Technologies) according to Poisson statistics (two-tailed t-test; *p<0.05, **p<0.001). (**D**) Immunofluorescence staining of lineage-specific markers KRT5 (basal) and KRT8/18 (luminal) in wild type, *Pbsn-Cre Gata3*f/f and *Krt5*CreERT2*Gata3*f/f allografts. Scale bar is representative of 50 µm. See also *Figure 1—figure supplements 1–2*.

*Figure 1 continued on next page*

*Figure 1 continued*

The online version of this article includes the following source data and figure supplement(s) for figure 1:

**Source data 1.** Statistical analysis for *Figure 1A–B* and *Figure 1—figure supplement 2A–B*.

**Source data 2.** Statistical analysis for *Figure 1C*.

**Figure supplement 1.** GATA3 is expressed in prostate basal and luminal cells.

**Figure supplement 1—source data 1.** Expression levels of differentially expressed genes between populations on *Figure 1—figure supplement 1C*.

**Figure supplement 2.** *Gata3* is important for propagation and differentiation of organoids.

**Figure supplement 2—source data 1.** Statistical analysis for *Figure 1—figure supplement 2F*.

**Figure supplement 3.** Ductal structures with multiple alveoli and a lumenized epithelial structure in allografts and organoids.

levels were unaffected by *Gata3* expression levels (*Figure 1—figure supplement 2C–D-E*), indicating that the stem/progenitor maintenance potential rather than cell proliferation or survival is altered in those organoids. To study the effect of *Gata3* loss on cell differentiation, testosterone was added to the media to favor differentiation of *Pbsn-Cre Gata3^{f/f}* prostate organoids which revealed a decrease in organoids capable of forming lumens, pointing to a cell differentiation defect associated with the increase in organoid-forming potential (*Figure 1—figure supplement 2F*; *Figure 1—figure supplement 2G*).

To confirm that the increased organoid-forming potential of *Gata3*-deficient cells is intrinsic to basal cells, we used the tamoxifen-inducible *Krt5^{CreERT2}* knock-in allele. In vitro activation of CreERT2 in KRT5+ cells reproduced the results obtained with the *Pbsn-Cre* strain (*Figure 1B* and *Figure 1—figure supplement 2B*). We further assessed the possibility that the increased organoid propagation potential comes from luminal cells generated during organoid growth and differentiation (*Figure 1—figure supplement 3A–B*). For this, we specifically inactivated *Gata3* in vitro in luminal cells using *Krt8^{CreERT2}* transgene. This did not lead to an increase in organoid-forming potential (*Figure 1B* and *Figure 1—figure supplement 2B*), indicating that the role of GATA3 in basal stem/progenitor cell homeostasis is restricted to the basal compartment.

We next tested the capacity of GATA3 to regulate stem/progenitor cell homeostasis in vivo by allograft assay. To this end, basal cells from wild type, *Pbsn-Cre Gata3^{f/f}* and *Krt5^{CreERT2} Gata3^{f/f}* prostates (all expressing *Rosa26^{TdTomato}* to trace donor tissues) were implanted with urogenital sinus mesenchyme (UGSM) (*Goldstein et al., 2010*; *Wang et al., 2009*) under the renal capsule of host mice and grown for 90 days in the presence of exogenous testosterone. Using serial dilution, we assessed the proportion of basal cells capable of generating TdTomato$^+$ prostate tissue [measured as prostate reconstituting units (PRU)] (*Figure 1C* and *Figure 1—figure supplement 3C*). The PRU frequency in the basal population was found to be 1 per 8600 for control. In contrast, *Pbsn-Cre Gata3^{f/f}* gave an average of 1 in 3,942, and *Krt5^{CreERT2}Gata3^{f/f}* of 1 in 1879 which corresponds respectively to over two- and four-fold increase in steady state stem/progenitor cell numbers in the *Gata3*-deficient basal cell pool. The difference between *Pbsn-Cre* and *Krt5^{CreERT2}* likely reflects the deletion efficiency of both transgenic lines in basal cells. The histological examination of these grafts showed ductal structures with multiple alveoli and a lumenized epithelial structure composed of a bilayer of basal and luminal cells as evidenced by KRT5 and KRT8/18 staining (*Figure 1D* and *Figure 1—figure supplement 3D*).

From these results, we conclude that GATA3 is critical for regulating basal stem/progenitor cell maintenance in the prostate. GATA3 inactivation promotes an increase in self-renewing capacity, leading to a gradual expansion of the stem/progenitor cell pool.

## BMP signaling is upregulated in Gata3-deficient organoids

To gain insight into the molecular mechanisms leading to the amplification of the stem cell pool upon *Gata3* inactivation, we performed RNA-seq on wild type and *Pbsn-Cre Gata3^{f/f}* prostate organoids harvested after 0, 2, 3 and 4 passages (*Figure 2A*). To track the deletion of *Gata3* exon four by the *Pbsn-Cre* transgene, we mapped RNA-seq transcript to the *Gata3* locus (*Figure 2B*). Surprisingly, cells from passage 0 mostly retained the floxed exon 4 of *Gata3*. However, loss of exon four occurred in about 50% of cells at passage 2, and in the vast majority of cells from passages 3 and 4. This finding, likely due to a limited efficiency of *Pbsn-Cre* in basal cells, allowed us to use the dynamics of locus deletion as an additional filter to identify GATA3 regulated genes.

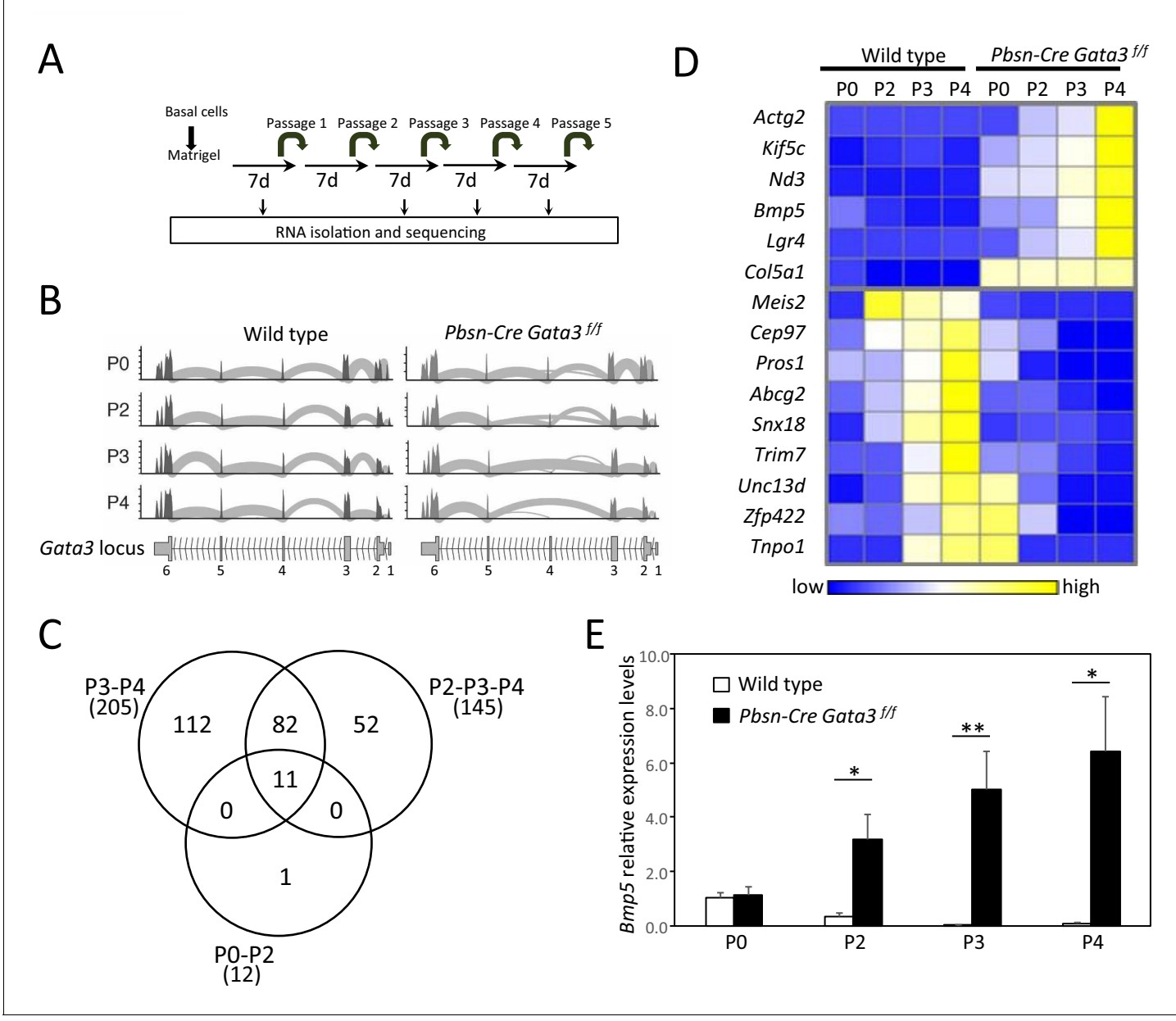

**Figure 2.** *Bmp5* expression in organoids is regulated by *Gata3*. (**A**) Schematic of RNA-seq strategy. mRNA was isolated from 4 days old wild type and *Pbsn-Cre Gata3*^f/f organoids at passages P0, P2, P3 and P4. (**B**) Deletion of exon four in *Pbsn-Cre Gata3*^f/f samples increases with passages. Shown are the read counts from RNAseq assigned to the *Gata3* locus in samples isolated from wild type and *Pbsn-Cre Gata3*^f/f prostate tissue at passage P0, P2, P3 or P4. (**C**) Venn diagram of genes differentially expressed between wild type and *Pbsn-Cre Gata3*^f/f prostate organoids using likelihood-ratio test with q-value <0.01. (**D**) Heatmap of log$_2$ transformed mRNA read counts of differentially expressed genes between wild type and *Pbsn-Cre Gata3*^f/f organoids and whose expression pattern follows *Gata3* loss with passages. (**E**) *Bmp5* mRNA expression levels as assessed by quantitative RT-PCR in both wild type and *Pbsn-Cre Gata3*^f/f organoids over passages. Data represent the average ± SD from three independent cDNA obtained from a pool of prostate cells from a minimum of three mice. Relative mRNA expression levels are normalized to *Ppia* mRNA levels (two-tailed t-test as compared to wild-type condition; *p<0.01, **p<0.005). See also *Figure 2—figure supplements 1–2*.

The online version of this article includes the following source data and figure supplement(s) for figure 2:

**Source data 1.** Expression levels of differentially expressed genes between wild type and *Pbsn-Cre Gata3*^f/f associated with *Figure 2C*.
**Source data 2.** Expression value for *Figure 2D*.
**Source data 3.** Statistical analysis for *Figure 2E*.
**Figure supplement 1.** *Gata3*-dependent gene signature.
**Figure supplement 1—source data 1.** Expression levels of differentially expressed genes between wild type and *Pbsn-Cre Gata3*^f/f displayed on *Figure 2—figure supplement 1A* and enrichment analysis from *Figure 2—figure supplement 1C*.
*Figure 2 continued on next page*

The comparison of differentially regulated genes between wild type and *Pbsn-Cre Gata3^f/f^* samples at passages 3 and 4 identified 205 candidate genes (*Figure 2C* and *Figure 2—figure supplement 1A*). Of interest, *Bmp5* emerged as the candidate with the highest differential expression ratio between wild type and mutant at passages 2, 3 and 4 (up to 48-fold difference), while being unaffected at passage 0, when the *Gata3* locus is still intact (*Figure 2D* and *Figure 2—figure supplement 1B*). A quantification of *Bmp5* levels by qRT-PCR validated the RNA-seq results (*Figure 2E*). In line with the dynamics of *Bmp5* expression, the analysis of all differentially expressed genes with the ENRICHR resource (*Chen et al., 2013*; *Kuleshov et al., 2016*) identified a number of genes regulated by SMAD, BMP, BMPR1a- and BMPR2 (Software ARCH2 and GEO datasets; *Figure 2—figure supplement 1C*) as well as a strong enrichment for SMAD4 binding sites in the regulatory region of those genes (Software Encode-ChEA; *Figure 2—figure supplement 1C*).

On the grounds of the consistent association between the GATA3 response in prostate stem/progenitor cells and the BMP/SMAD signaling pathway, we decided to probe further the role of BMP5 in prostate stem cell regulation.

## BMP signaling regulates basal stem/progenitor cell maintenance

Previous studies have linked canonical BMP signaling to progenitor cell proliferation and differentiation in the epidermis, hair follicle and intestine (*Genander et al., 2014*; *Haramis et al., 2004*; *He et al., 2004*; *Lewis et al., 2014*). However, relatively little is known about the regulation of stem cell homeostasis by BMPs, notably in the prostate. To better understand the source of BMP5 in the adult prostate, we measured *Bmp5* expression by qRT-PCR in basal, luminal and stromal cells sorted from wild type and *Pbsn-Cre Gata3^f/f^* prostates (*Figure 2—figure supplement 2A*). In wild-type prostates, *Bmp5* was found to be expressed in all three cell types (*Figure 2—figure supplement 2A*). Interestingly, *Gata3* inactivation primarily increased *Bmp5* expression in basal cells, which suggests a specific role for *Gata3* in this compartment. We next sought to determine whether GATA3 directly binds the *Bmp5* locus. For this, we first probed ChIP-seq data from the Gene transcription regulation database (GTRD), which identified several regions of the *Bmp5* locus bound by GATA3 in murine and human cells (*Figure 2—figure supplement 2B*). To validate this possibility in prostatic tissue, we took advantage of a biotinylated allele of *Gata3* (*Gata3^bio^*) and performed BioChIP-PCR assay on adult prostates. Using this high-affinity system, we found that GATA3 is bound to the *Bmp5* locus which suggest that GATA3 regulates directly *Bmp5* expression in prostate basal cells (*Figure 2—figure supplement 2C*). In order to clarify the genetic relationship between *Gata3* and *Bmp5*, we used the 'small ear' (SE) mouse strain, which harbor a nonsense mutation in the propeptide region of *Bmp5* that prevents mature protein expression (*King et al., 1994*; *Figure 3—figure supplement 1A*). We found no difference in *Gata3* expression levels in basal cells in the absence of *Bmp5* both by qRT-PCR and by FACS using *Gata3^GFP^* knock-in reporter mouse strain on a wild type or *Bmp5^SE/SE^* background (*Figure 3—figure supplement 1B–C*), suggesting that BMP5 is not a critical regulator of *Gata3* expression in the prostate.

To test whether BMP signaling affects the self-renewal potential of *Gata3*-deficient basal cells, we blocked it with the inhibitory protein NOGGIN in culture. In wild-type organoids, NOGGIN treatment prevented basal cells from being passaged past four generations (*Figure 3A* and *Figure 3—figure supplement 2A*). Strikingly, NOGGIN treatment also abrogated the long-term amplification of *Gata3*-deficient organoids, suggesting that BMP signaling is a key mediator of the *Gata3*-deficient propagation phenotype (*Figure 3A* and *Figure 3—figure supplement 2A*). This decrease in organoid-forming potential was not caused by changes in proliferation nor apoptosis, as shown by unaltered Ki67 and TUNEL stainings in treated organoids (*Figure 3—figure supplement 2E–F*). To validate these result, we used K02288, a potent small molecule inhibitor selective to BMPR-SMAD1/

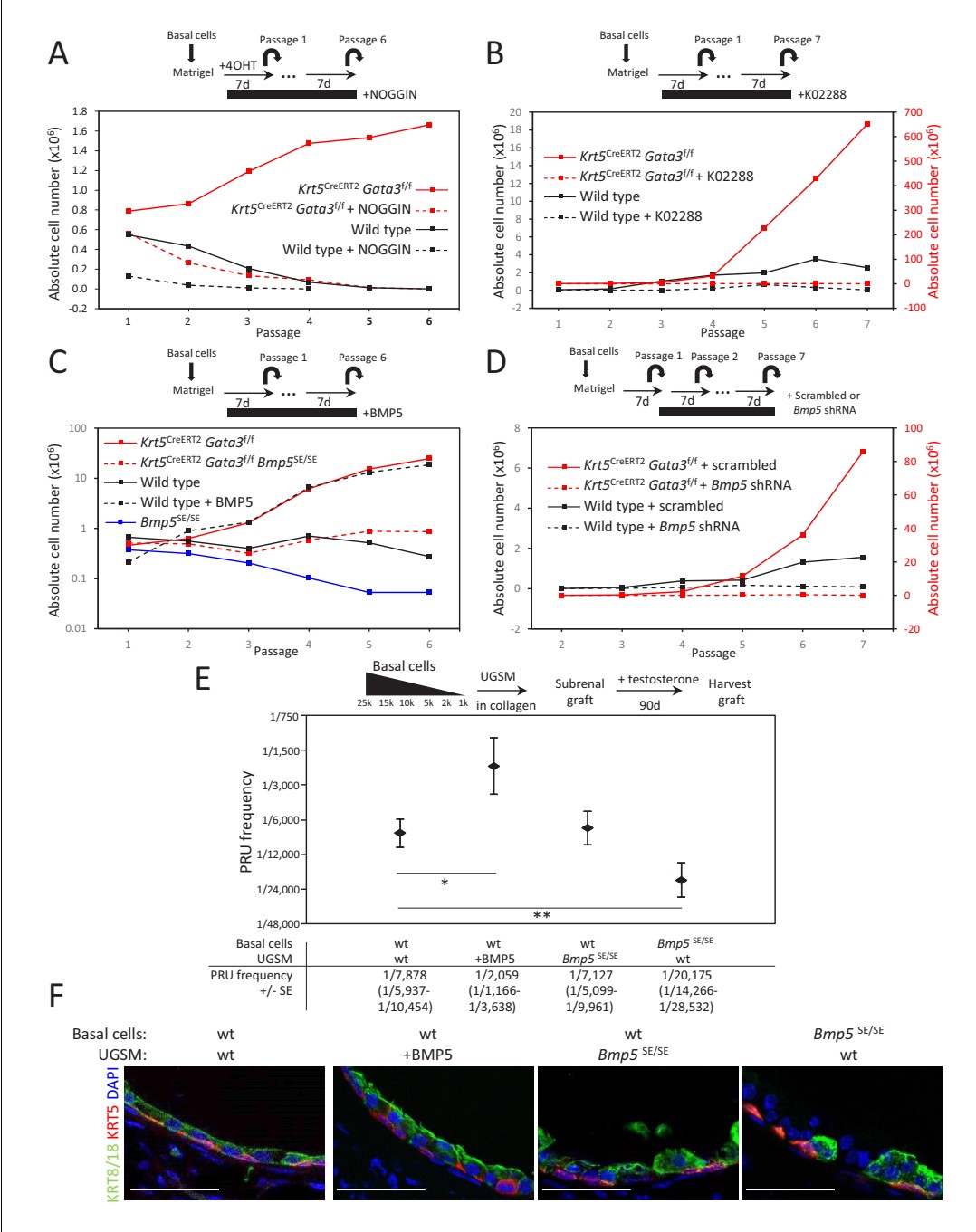

**Figure 3.** BMP5 increases the propagation potential of basal stem/progenitor cells. (**A–B**) Organoid-forming capacity is abrogated by both BMP and BMPR-SMAD1/5/8 inhibitors treatment. Sorted basal cells from wild type or $Krt5^{CreERT2}Gata3^{f/f}$ prostate were grown in presence or absence of NOGGIN (**A**) or small inhibitor K02288 (**B**) for six passages. Cre activity was induced by hydroxy-tamoxifen treatment in vitro for the first passage (**A**) or by tamoxifen injection in vivo 4 weeks prior to culture (**B**). Organoid-forming potential was assessed as in *Figure 1A*. (**C**) *Bmp5* loss reduces organoid-forming activity in vitro, while propagation potential capacity of wild type cells is increased by BMP5 treatment. Basal cells ($10^5$) from wild type, $Bmp5^{SE/SE}$, $Krt5^{CreERT2}Gata3^{f/f}$ or $Krt5^{CreERT2}Gata3^{f/f}$ $Bmp5^{SE/SE}$ mice were grown for six passages. Exogenous BMP5 was added to culture media where indicated. Cre activity was induced in vivo as in (**B**). (**D**) *Bmp5* silencing specifically affects organoid-forming activity in vitro. ShRNA against *Bmp5* or scrambled were electroporated in first passage organoid from wild type and $Krt5^{CreERT2}Gata3^{f/f}$ basal cells and grown for seven passages. Cre activity was induced in vivo as in (**B**). (**E**) *Bmp5* levels affects regenerative potential in vivo. Different numbers of sorted basal prostate cells from $Rosa26^{LTdTomato}$ (control) or $Bmp5^{SE/SE}Rosa26^{LTdTomato}$ were transplanted with UGSM either wild type, ectopically expressing *BMP5* or derived from $Bmp5^{SE/SE}$ mice. Limiting dilution analysis was done as in *Figure 1C*. Notice that the loss of *Bmp5* in basal cells but not in UGSM affects prostate reconstituting units (PRU) frequency (two-tailed t-test as compared to wild-type condition; **p<0.02, *p<0.04). (**F**) Immunofluorescence of allografts
*Figure 3 continued on next page*

*Figure 3 continued*

show presence of prostatic ducts expressing both KRT5 and KRT8/18. Scale bar is representative of 50 µm. See also *Figure 3—figure supplements 1–2*.

The online version of this article includes the following source data and figure supplement(s) for figure 3:

**Source data 1.** Statistical analysis for *Figure 3A–D* and *Figure 3—figure supplement 2A–D*.
**Source data 2.** Statistical analysis for *Figure 3E*.
**Figure supplement 1.** GATA3 expression is not regulated by *Bmp5* expression in the prostate.
**Figure supplement 1—source data 1.** Expression levels of *Gata3* between wild type and *Bmp5*$^{SE/SE}$ associated with *Figure 3—figure supplement 1B*.
**Figure supplement 2.** BMP5 treatment does not affect proliferation nor survival in organoid.
**Figure supplement 3.** Allografts form well-organized ductal structures with both basal and luminal lineages.

5/8 signaling (*Sanvitale et al., 2013*) which showed similar results to NOGGIN treatment (*Figure 3B* and *Figure 3—figure supplement 2B*).

To determine whether BMP5 is sufficient to promote basal stem/progenitor cell maintenance, we treated wild-type basal prostate organoids with exogenous BMP5 over several passages. As expected, BMP5 treatment increased the organoid-forming capacity of basal cells over time (*Figure 3C* and *Figure 3—figure supplement 2C*). Conversely, *Bmp5*$^{SE/SE}$ cells showed an impaired organoid-forming potential comparable to NOGGIN or K02288-treated cells (*Figure 3C* and *Figure 3—figure supplement 2C*). Here again, neither survival (*Figure 3—figure supplement 1D*) nor proliferation was affected in the organoids that successfully grew, which supports a self-renewal phenotype. We next assessed whether *Gata3*-deficient phenotype was specifically dependent on the BMP5 ligand by inactivating both *Bmp5* and *Gata3* in KRT5-positive basal cells in vivo. *Bmp5* mutation impaired the increased organoid-forming capacity of *Gata3*-deficient cells (*Figure 3C* and *Figure 3—figure supplement 2C*). In addition, the acute loss of *Bmp5* using shRNA against *Bmp5* showed a similar effect, inhibiting the passaging capacity of both wild type and *Gata3*-deficient organoid (*Figure 3D* and *Figure 3—figure supplement 2D*). Together, these results confirm that the increased propagation potential of *Gata3*-deficient basal cells requires BMP signaling driven by the BMP5 ligand.

To validate this phenotype in vivo, we assessed whether BMP5 is sufficient to promote prostate tissue development from basal stem/progenitor cells in an allograft assay. For this, we compared the regenerative potential of wild-type basal cells (expressing the constitutively active *Rosa26*$^{TdTomato}$ lineage tracer) embedded in either wild-type UGSM or UGSM engineered to overexpress BMP5 (*Figure 3E*). After 90 days of growth under the kidney capsule, the PRU frequency averaged 1 in 7878 in normal UGSM and increased to 1 in 2059 in BMP5-expressing UGSM, an effect equivalent to the loss of *Gata3* in KRT5+ basal cells (1 in 1,879) (*Figure 1C*). These grafts showed a formation of well-organized ductal structures including both basal and luminal lineages (*Figure 3F* and *Figure 3—figure supplement 3*). Hence, the exposure of basal stem/progenitor cells to increased BMP5 levels is sufficient to increase their regenerative potential. However, this experiment does not directly address whether BMP5 acts as a maintenance factor from the stromal mesenchyme or from within the basal cell compartment. To test this, we compared the regenerative potential of wild-type basal cells embedded in mesenchyme derived from either wild-type or *Bmp5*$^{SE/SE}$ animals (*Figure 3E*). This experiment revealed no significant difference in the PRU frequency or lineage potential of the grafts grown in presence or absence of mesenchymal *Bmp5*, indicating a role for BMP5 as a regulator of stem/progenitor cell homeostasis intrinsic to the basal cell layer.

To directly assess the autonomous role of BMP5 in the basal compartment, we compared the regenerative potential of wild type or *Bmp5*$^{SE/SE}$ basal cells (expressing the constitutively active *Rosa26*$^{TdTomato}$ lineage tracer) embedded in wild-type UGSM cultures. After 90 days under the kidney capsule, the calculated PRU frequency averaged 1 in 7878 in the wild-type basal population but dramatically decreased to 1 in 20,175 in the absence of *Bmp5* (*Figure 3E*). Together, these results demonstrate that BMP5 is critically required to sustain a full regenerative capacity in basal cell-derived allografts and identify BMP5 as an important regulator of basal stem/progenitor cells maintenance in the prostate.

## BMP5 deficiency delays Pten-dependent tumor progression

One of the earliest and most frequent events in prostate cancer progression is the loss of the tumor suppressor PTEN (*Abeshouse et al., 2015*). Accordingly, *Pten* loss in the mouse prostate leads to carcinoma within 6–8 weeks (*Mulholland et al., 2011*; *Trotman et al., 2003*; *Wang et al., 2003*). Those tumors are castration-resistant, which mimics recurrent castration-resistant prostate cancer (CRPC) and are highly enriched in prostate stem cells (*Wang et al., 2003*). We previously showed that GATA3 is progressively lost in the prostate epithelium of *Pten*-deficient mice, while enforced GATA3 expression slows down tumor progression (*Nguyen et al., 2013*). In light of these results, we hypothesized that BMP5 signaling may contribute to the maintenance of *Pten*-deficient cancer stem cells.

To assess this possibility, we first performed a short-term organoid-forming assay using purified basal cells from $Krt5^{CreERT2}Pten^{f/f}$ adult prostates. As expected, in vitro induction of the Cre recombinase by tamoxifen treatment led to an increase in organoid-forming potential over time (*Figure 4A* and *Figure 4—figure supplement 1A*). To test whether BMP signaling affects the self-renewal potential of cancer cells, we treated *Pten*-deficient basal prostate organoids with the inhibitory protein NOGGIN. Inhibition of BMP signaling affected their organoid-forming capacity, abrogating the long-term expansion in organoid cultures (*Figure 4A* and *Figure 4—figure supplement 1A*). This result suggests that the increased propagation potential of *Pten*-deficient prostate cancer stem/progenitor cells also requires BMP signaling.

To validate this result, we used the specific BMPR-SMAD1/5/8 signaling inhibitor K02288 on two *Pten*-deficient cell lines (CaP2 and CaP8). K02288 specifically abrogated culture growth without affecting cell viability or the expression or phosphorylation of the PTEN effector AKT (*Figure 4—figure supplement 2A–B*), indicating that K02288 acts downstream of AKT-mediated signaling.

We then tested BMP signaling inhibition in vivo by K02288 treatment of $Krt5^{CreERT2}Pten^{f/f}$ adult mice for 30 days (at which point the mock-treated mice developed skin tumors and had to be sacrificed) (*Figure 4B–E*). Drug treatment led to a reduction of prostate lesions in comparison to mock-treated mice that had accumulated prostatic intraepithelial neoplasia (PIN) at this stage (*Figure 4C*). Interestingly, K02288 treatment additionally led to a marked reduction of the severity of skin lesions derived from KRT5-positive basal cells and greatly delayed the onset of tumor growth (*Figure 4D-E* and *Figure 4—figure supplement 1A*), indicating that the promotion of *Pten*-deficient tumor expansion by BMP signaling is not limited to the prostate. To determine whether *Pten*-deficient neoplasia was specifically dependent on the BMP5 ligand, we inactivated both *Pten* and *Bmp5* in KRT5-positive basal cells of the prostate and skin (*Figure 4G*). Loss of *Bmp5* impaired the aberrant organoid-forming capacity of *Pten*-deficient prostate cells (*Figure 4F* and *Figure 4—figure supplement 1B*), as well as the increase in numbers of aberrant SCA1$^{hi}$ luminal progenitor cells (*Figure 4K–L*). Strikingly, $Krt5^{CreERT2}Pten^{f/f}Bmp5^{SE/SE}$ mice showed a reduction in the formation of prostate and skin hyperplasia as well as a delay in tumor growth onset similar to animals treated with K02288 (*Figure 4H,I,J* and *Figure 4—figure supplement 1D, E, F*). This correction of prostate tumor phenotype by *Bmp5* loss was still observed 9 weeks after Cre induction (*Figure 4M*). Interestingly, prostate and skin tumor phenotypes could also be dampened by overexpression of *GATA3* in these tissues (*Figure 4H–L*) indicating that both *Gata3* and BMP5 are key players in *Pten*-deficient cancer progression. These results additionally raise the possibility to use a small inhibitor against BMP signaling as a treatment against PTEN-deficient cancer progression both in the prostate and in the skin.

## Discussion

In recent years, the presence of adult prostate stem cells has been reported in both the basal and luminal compartments (*Choi et al., 2012*; *Goldstein et al., 2010*; *Liu et al., 2011*; *Wang et al., 2013*). However, the question of how these cells regulate the balance between long-term maintenance and differentiation remained unanswered. Here, we tackle this question in adult basal stem/progenitor cells. We show that prostate basal cells devoid of the transcription factor *Gata3* increase their self-renewal capacity in long-term propagation assays and further identify BMP5 as a crucial mediator of this activity. Using gain and loss-of-function approaches, we demonstrate that BMP signaling is required for sustained stem/progenitor cell propagation within the basal cell compartment. Using a mouse model of prostate cancer highly enriched in cancer stem cells, we finally show that inhibition of BMP5 signaling is sufficient to delay cancer progression from basal cells in the prostate

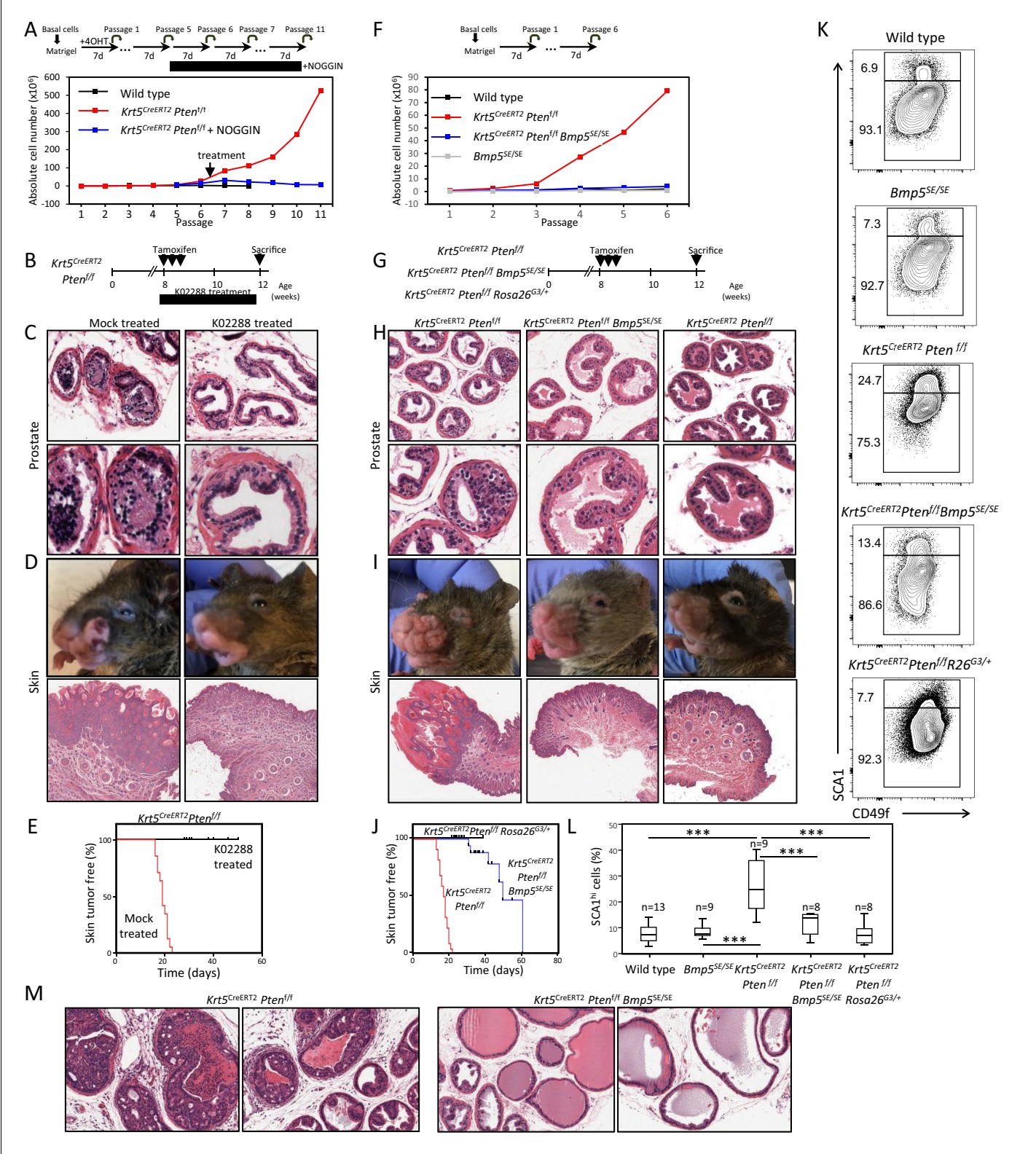

**Figure 4.** BMP inhibition reduces *Pten*-deficient propagation potential and inhibits skin and prostate tumor initiation. (**A**) The aberrant organoid-forming capacity of *Pten*-deficient basal cells is abrogated by BMP inhibitor (NOGGIN) treatment. Cre activity was induced in vitro by treatment with hydroxy-tamoxifen for the first passage of organoid derived from basal cells of *Krt5^CreERT2^Pten^f/f^* mice. From passage 5, organoids were cultured in presence or absence of NOGGIN. Organoid-forming potential was assessed as in ***Figure 1A***. (**B**) *Krt5^CreERT2^Pten^f/f^* tamoxifen-treated mice were

*Figure 4 continued on next page*

*Figure 4 continued*

injected with either K02288 or PBS for 4 weeks. (**C–D**) Representative histological sections of prostate tissue (**C**) and skin tissue (**D**) stained with H&E showing an absence of PIN and skin hyperplasia in K02288-treated as compared to mock-treated mice. (**E**) Kaplan-Meier skin tumor free survival curves of tamoxifen-induced $Krt5^{CreERT2}Pten^{f/f}$ treated or not with K02288 (log-rank (Mantel-Cox) test; p<0.0001, n = 7 and n = 14, respectively). Tick-marks represent sacrificed animals. (**F**) $Bmp5$ loss rescues the aberrant organoid-forming capacity of $Pten$-deficient basal cells. Cre activity was induced in vivo by tamoxifen injection in adult mice 4 weeks prior to organoid propagation potential assessment. (**G–L**) Eight-week-old mice were treated with tamoxifen and sacrificed 4 weeks later. (**H–I**) $Bmp5$ loss and $Gata3$ overexpression rescues $Pten$-deficient prostate and skin hyperplasia. Shown are representative H&E pictures of prostate (**H**) and skin (**I**) tissues. (**J**) Kaplan-Meier skin tumor free survival curves of tamoxifen-induced $Krt5^{CreERT2}Pten^{f/f}$, $Krt5^{CreERT2}Pten^{f/f}Bmp5^{SE/SE}$ and $Krt5^{CreERT2}Pten^{f/f}Rosa26^{G3/+}$ mice (log-rank (Mantel-Cox) test; p<0.0001, n = 22, n = 17 and n = 9, respectively). Tick-marks represent sacrificed animals. (**K–L**) Representative FACS phenotype (**K**) and percentage of SCA1$^{hi}$ cells (**L**) in the luminal compartment as defined by Lin(CD45, CD31, TER119)⁻EpCAM⁺CD49f$^{Med}$ of total prostate (one-way ANOVA; ***p<0.0001). (**M**) Representative H&E pictures of prostate tissues of $Krt5^{CreERT2}Pten^{f/f}$ and $Krt5^{CreERT2}Pten^{f/f}Bmp5^{SE/SE}$ mice sacrificed 9 weeks after tamoxifen treatment. See also *Figure 4—figure supplements 1,2*. The online version of this article includes the following source data and figure supplement(s) for figure 4:

**Source data 1.** Statistical analysis for *Figure 4A–F* and *Figure 4—figure supplement 2A*; *Figure 4—figure supplement 2B*.
**Source data 2.** Statistical analysis for *Figure 4E–J*.
**Source data 3.** Statistical analysis for *Figure 4L*.
**Figure supplement 1.** BMP inhibition corrects *Pten*-deficient tumor phenotypes.
**Figure supplement 2.** K02288 acts downstream of AKT pathway.
**Figure supplement 2—source data 1.** Statistical analysis for *Figure 4—figure supplement 2A*.
**Figure supplement 2—source data 2.** Full unedited gels for *Figure 4—figure supplement 2B*.

and in the skin. Together, these observations demonstrate that BMP signaling promotes basal stem cell self-renewal, while GATA3 counteracts this activity to regulate the basal stem cell pool.

BMP signaling has been linked to stem cell maintenance in a number of organisms, often acting as a niche factor (*Chen et al., 2011*; *Oshimori and Fuchs, 2012*; *Tian and Jiang, 2014*). However, whether BMP signaling promotes stem cell self-renewal or quiescence is context-dependent (*Badeaux et al., 2013*; *Chen et al., 2011*; *Chung et al., 2018*; *Genander et al., 2014*; *Kangsamaksin and Morris, 2011*; *Lewis et al., 2014*; *Li et al., 2012*; *Oshimori and Fuchs, 2012*; *Tadokoro et al., 2016*; *Tian and Jiang, 2014*). Our results in the prostate clearly identify a role for BMP signaling in promoting the long-term maintenance of prostate stem/progenitor cells, that appears to be intrinsic to the basal cell compartment as opposed to an exogenous niche factor. In support of this, *Gata3* deficiency leads to an increase in BMP5 expression specifically in basal cells, while *Gata3*-deficient prostate organoids show a strong stem/progenitor cell amplification phenotype despite being devoid of stromal niche. This basal cell amplification can be blocked by BMP inhibitor treatment and is blunted in *Bmp5*-deficient cells indicating that BMP signaling is required for stem/progenitor cell amplification. An alternative possibility would be that BMP5 acts as a niche factor from luminal cells. However, *Gata3* inactivation by luminal-specific $Krt8^{CreERT2}$ failed to increase the organoid-forming potential, arguing against this possibility. Importantly, while BMP5-overexpressing mesenchyme could enhance basal stem cell activity in renal allograft assay, *Bmp5*-deficient mesenchyme remained competent in stem cell derived allograft growth. In contrast, *Bmp5*-deficient basal cells had a blunted regenerative potential in the presence of wild-type mesenchyme. Hence, increased exposure to BMP5 expressed from the UGSM can support basal stem/progenitor cells, but ultimately it is BMP5 expression from the basal compartment that is critical to basal stem cell homeostasis. The intrinsic role of BMP5 in the basal cell compartment is further supported by the identification of *Bmp5* as one of the most highly expressed genes in a prostate basal stem cell population defined by s-SHIP1 gene expression (*Brocqueville et al., 2016*). The molecular mechanism by which GATA3 modulates *Bmp5* in basal cells remains elusive. However, the direct binding of GATA3 to the *Bmp5* locus in mouse and human, combined with the fact that GATA3 is known to interact with repressive chromatin modifier complexes (*Tremblay et al., 2018*) suggests a molecular mechanism to be explored further.

Taken together, these results favor a model by which GATA3 regulates stem/progenitor cell long-term maintenance by modulating the expression levels of the autocrine stem cell factor BMP5.

PTEN mutation is one of the earliest and most frequent alterations in prostate cancer (*Abeshouse et al., 2015*). Accordingly, *Pten*-deficient prostates develop castration-resistant prostate cancer (*Mulholland et al., 2011*; *Wang et al., 2003*) that are highly enriched in cancer

stem cells (this report) (*Goldstein et al., 2010*; *Mulholland et al., 2009*; *Wang et al., 2014a*). To validate whether the increase in BMP signaling was involved in *Pten*-deficient tumors, we treated *Pten*-deficient prostate organoids and mouse prostate tumors with a selective inhibitor against BMP signaling, which resulted in blunted organoid passaging potential and a delay in tumor progression in vivo. These results are in line with a previous report showing that loss of TGFβ signaling has been linked to prostate cancer progression in *Pten*-deficient tumor, in part through upregulation of BMP signaling (*Zhao et al., 2018*). Here, we identify the ligand BMP5 as prime driver of *Pten*-deficient cancer progression. It is possible that other BMPs are involved in tumor progression that would also be blocked by small molecule-mediated inhibition of BMP signaling. For example, BMP6, a member of the BMP5/6/7 subfamily, is upregulated in prostate cancer and contributes to cancer progression (*Darby et al., 2008*). Whether BMP5 and BMP6 have overlapping or distinct roles remains to be determined. However, our results clearly demonstrate that *Bmp5* inactivation alone is sufficient to significantly delay cancer development with an efficiency similar to BMP pathway inhibition, identifying this ligand as a central player in the process.

In this study, we specifically inactivated *Pten* in basal cells, that are known to transition to a luminal fate and generate carcinoma under stress conditions (*Wang et al., 2014b*; *Wang et al., 2013*). Our results demonstrate that dampening the BMP-regulated cancer stem cell pool in basal cells is an effective strategy to reduce the tumorigenic potential of this population. Whether or not the GATA3-BMP axis is also effective in luminal cells of the prostate is still unclear. Our results suggest that the upregulation of BMP5 by *Gata3* loss occurs mostly in basal cells. However, the fact that *Gata3* could modulate prostate cancer progression in *Pbsn-Cre; Pten*-deficient mice (*Nguyen et al., 2013*) known to develop largely from luminal cells raise the possibility of a role in luminal cells. One possibility would be that *Gata3* acts through the regulation of other BMP family members to achieve a similar role in different cell types.

Remarkably, as skin cancer can originate from KRT5-positive basal stem cells (*Suzuki et al., 2003*), the strategy of BMP inhibition proved also effective in this tissue. BMP5 was previously reported to play a role in non-cancerous keratinocytes where *Bmp5* inactivation reduces the number of label-retaining cells, while exogenous BMP5 increased colony formation (*Badeaux et al., 2013*; *Kangsamaksin and Morris, 2011*). These results support a role for BMP5 in stem cell maintenance, comparable to the one described here. The effective activity of BMP inhibition in both tissues further raise the prospect of a therapeutic approach in the treatment of *Pten*-deficient tumors.

## Materials and methods

### Mice

All experimental mice were kept in a C57BL/6 background. *Pbsn-Cre* (*Wu et al., 2001*), *Gata3*^flox, *Gata3*^GFP, *Rosa26*^GATA3 (*Grote et al., 2006*), *Gata3*^bio (kind gift from Dr. Busslinger, IMP, Vienna; see *Figure 2—figure supplement 3* for generation strategy), *Rosa26*^BirA (*Wood et al., 2016*), *Krt5*^CreERT2 (*Van Keymeulen et al., 2011*), *Krt8*^CreERT2 (*Choi et al., 2012*), *Rosa26*^LstopLTdTomato (*Madisen et al., 2010*), *Pten*^flox (*Trotman et al., 2003*) and *Bmp5*^SE (*Kingsley et al., 1992*) mice were described previously. Constitutively active *Rosa26*^LTdTomato were generated from female *Pbsn-Cre* mice which express Cre in the germline and backcrossed to C57BL/6 to eliminate the Cre allele. Immunodeficient SCID-beige mice were obtained from Charles River and kept in pathogen-free conditions. Except stated otherwise, all experiments were done using 8-week-old adult mice. In vivo CreERT2 induction was done by intraperitoneal injection of three doses of 3 mg of tamoxifen dissolved in corn oil. Mice were injected daily intraperitoneally with either mock or 500 mg of K02288 (AdooQ Bioscience) dissolved in PBS containing 10% DMSO. Kaplan–Meier skin tumor-free survival curves were obtained using Prism 6.0 software (GraphPad). All animal procedures were approved by McGill University Animal Care Committee according to the Canadian Council on Animal Care guidelines for use of laboratory animals in biological research.

### FACS sorting and analysis

Whole prostate tissue from a minimum of three mice were dissected and pooled in cold PBS 2% FBS, minced and digested at 37°C for 2 hr in DMEM 5% FBS 1X Collagenase/hyaluronidase (Stemcell Technologies) for 5 min in 0.25% trypsin/EDTA and followed by 10 min in dispase II (5 U/ml) and

DNase I (0.1 mg/ml) (Roche). All solution contained 10 µM of Rock inhibitor Y-27632 (ApexBio). The digested cells were passed through a 27G needle and filtered through a 70 µm cell strainer. Cell staining was done on ice for 30 min using human IgG blocking solution and antibodies CD45 (30-F11), TER119 (TER-119), CD31 (MEC13), CD49f (GoH3), EpCAM (G8.8), SCA1 (D7) (all from Biolegend) and TROP2 (from R and D). Dead cells were excluded by fixable Viability dye (eBioscience) staining. FACS analysis and sorting was performed on a BD Fortessa and Aria Fusion apparatus (BD Biosciences). Data was analyzed using DIVA (BD Biosciences) and FlowJo softwares.

## Short-term organoid propagation assay

$LIN^-SCA1^+CD49F^+EPCAM^+TROP2^+$ basal cells were sorted from whole prostate and plated in advanced DMEM/F12 (ThermoFisher) 50% matrigel (Corning) around the rims of 12-well plates. Organoids were grown for 7 days in modified organoid media composed of WIT basic media (Cellaria) supplemented with 10 ng/ml EGF, 25 ug/ml BPE, 1X B27 (Life Technologies) and 10 µM Y-27632. Pictures of organoid culture were taken with an AxioObserver (Zeiss) and organoid size was analyzed using ImageJ (Fiji) software. Passaging of organoids was done by digestion with dispase for 1 hr at 37°C and Trypsin 0.25% EDTA for 5 min, syringe trituration and replating in 50% matrigel. Viable cells from dissociated organoids were counted using TC10 cell counter (Biorad) with trypan blue and replated at a concentration of $10^5$ per wells. In vitro CreERT2 induction was done by treatment of organoid with 500 ng/ml 4-hydroxy-tamoxifen (4-OHT). Media supplemented with either exogenous 50 ng/ml of recombinant hBMP5 (Aviscera Bioscience), 180 ng/ml recombinant hNOGGIN (RayBiotech) or 10 µM of K02288 inhibitor (AdooQ Bioscience) was changed every other day. Organoid differentiation was done by supplementing culture media with $10^{-8}$M 5α-dihydrotestosterone (DHT, Steraloids). Electroporation of shRNAs either scrambled or against *Bmp5* (MISSION pLKO.1-Puro shRNA; Sigma-Aldrich) was done on single cells dissociated from passage one organoid using P1 Primary Cell 4D-Nucleofector Kit (Bioscience) using program EL-110 and selected using 0.5 µg/ml of puromycin. Growth rate of cells upon organoid passage was calculated using non-linear regression curve fitting and significance between genotype was assessed by one-way ANOVA using Prism 6.0 software (GraphPad).

## Allograft/limiting dilution assay

Urogenital sinus mesenchyme (UGSM) culture were described previously (*Xin et al., 2003*), briefly the urogenital sinus from wild type or *Bmp5^SE/SE^* embryos between E14.5 to E16.5 was digested in 1% trypsin in HBSS for 90 min at 4°C. Mesenchyme was separated from the epithelium and digested in 0.2% collagenase A for 1 hr at 37°C. Single cells were plated on fibronectin treated plastic in DMEM medium 5% FBS 5% Nu serum 1% Non-Essential Amino Acids (NEAA) 1% glutamine and 1 nM 5α-dihydrotestosterone (DHT, Steraloids). UGSM overexpressing BMP5, was generated by infection with lentiviral particles from either empty or BMP5 expressing pLenti plasmid and subjected to selection with puromycin. Subrenal regeneration assays have been previously described (*Doles et al., 2005*). Briefly, sorted TdTomato^+^ basal cells from a pool of a minimum of three mice from either *Pbsn-Cre Rosa26^LstopLTdtomato^*, *Pbsn-Cre Gata3^f/f^Rosa26^LstopLTdtomato^*, *Krt5^CreERT2^Gata3^f/f^Rosa26^LstopLTdtomato^*, constitutively active *Rosa26^LTdTomato^* or *Bmp5^SE/SE^Rosa26^LTdTomato^* genotype were mixed with $10^5$ UGSM cells of the indicated genotype in rat collagen I (Corning) and implanted under the kidney capsule of SCID-beige recipient mice. Growth of prostate tissue grafts was stimulated by subcutaneous implantation of silastic testosterone implant of 25 mg every 3 weeks for 90 days. An outgrowth was defined as an epithelial TdTomato^+^ fluorescent structure comprising ducts. A minimum of four replicates was done per dilution and using Poisson distribution, Prostate reconstituting unit (PRU) frequency was calculated for basal cells from the whole prostate following limiting dilution transplantation using L-calc software (Stem cell technologies). Student's t-tests were performed for statistical analysis between two groups.

## Microscopy and image analysis

Immunofluorescence and H&E staining were performed on PFA-fixed organoid or tissue embedded in paraffin or on freshly frozen tissue embedded in OCT, sectioned to obtain 4- and 10-µm-thick sections as described previously, respectively (*Nguyen et al., 2013*). Staining was done overnight using the following primary antibodies: KRT5 (1:500, Biolegend #905901), KRT8/18 (1:200, Fitzgerald

#20R-CP004), GATA3 (1:100, SantaCruz #sc-9009) and Ki67 (1:200, eBioscience #14-5698-82). TUNEL staining was done using the In Situ Cell Death Detection Kit (Roche), following manufacturer's protocol. Immunofluorescence images were acquired using either an LSM710, LSM780 or LSM800 confocal microscope (Zeiss). H&E images were scanned on an AperioScanScope AT, whereas brightfield images were taken with an Axio Observer Z1 (Zeiss).

## RNA isolation, quantitative RT-PCR and transcriptomic profiling

Total RNA was extracted from sorted population of prostate cells or organoid cultures using a RNeasy micro kit (Qiagen). To increase the proportion of stem/progenitor cells in the population, organoids were harvested after 4 days in culture (instead of 7). RNA was reverse transcribed with MMLV (Invitrogen) according to manufacturer's procedures. Real-time quantitative PCR was performed using Green-2-go mastermix (BioBasic Inc) on Realplex2 Mastercycler (Eppendorf). Total RNA was isolated and sequenced from day 4 organoids of wild type and *Pbsn-Cre Gata33^{f/f}* at passages 0, 2, 3 and 4. Sequencing libraries were prepared by Genome Quebec Innovation Centre (Montreal, Canada), using the TruSeq Stranded Total RNA Sample Preparation Kit (Illumina TS-122–2301, San Diego, CA) following depletion or ribosomal and fragmented RNA. The libraries were sequenced using the Illumina HiSeq 2000 sequencer, 100 nucleotide paired-end reads, generating approximately 60 million reads per sample. The sequencing reads were pseudo aligned to the mouse reference genome (mm10) using default parameters in Kallisto (*Bray et al., 2016*). Transcripts were annotated using Ensembl release 89. Abundance estimate and bootstrap values generated by Kallisto were used for expression quantification (*Bray et al., 2016*). Differential expression testing was performed to identify genes differently expressed between genotypes and passages with Sleuth using either a likelihood ratio test (LRT) or Walds test (WT) (*Pimentel et al., 2017*) from Sleuth package. Maps of sequencing reads were generated using R and the ggplot2 package (version 1), and bedtools (*Quinlan and Hall, 2010*). Analysis of *Gata3*-deletion of exon four by *Pbsn-Cre* was done by mapping raw reads to the *Gata3* genomic locus using the bedtools package. Heatmap of normalized gene expression (Average = 0; variance = 1) were generated using matrix2png program (*Pavlidis and Noble, 2003*). All raw and processed transcriptome data are available from NCBI GEO (accession GSE155289).

## Biotin-chromatin immunoprecipitation pulldown

Prostate tissue from *Gata3^{bio/bio} Rosa26^{BirA/BirA}* mice were minced down and crosslinked for 10 min with 1% formaldehyde. Formaldehyde was quenched by addition of 0.125 M glycine. Fixed cells were pelleted by centrifugation, washed twice in cold phosphate-buffered saline, then washed once in Triton buffer for 15 min (10 mM Tris-HCl, pH 8.0, 10 mM EDTA, 0.5 mM EGTA, 0.25% Triton X-100) and once in NaCl buffer for 15 min (10 mM Tris-HCl, pH 8.0, 1 mM EDTA, 0.5 mM EGTA, 200 mM NaCl). Cells were pelleted, resuspended in RIPA buffer (10 mM Tris-HCl, pH 8.0, 140 mM NaCl, 1 mM EDTA, 1% Triton X-100, 0.1% SDS, 0.1% deoxycholate) and sonicated (7 $\times$ 10 s bursts) to make soluble chromatin ranging in size from 500 to 1000 bp. Cellular debris were removed by centrifugation (16,000 $\times$ *g* for 10 min), and protein concentrations were determined by Bradford staining. Crosslinked extracts were subjected to BioChIP pulldown using streptavidin conjugated beads or IgG control beads. Bound extracts were sequentially washed twice with 1 ml of RIPA buffer, twice with 1 ml of LiCl buffer (10 mM Tris-HCl, pH 8.0, 250 mMLiCl, 1 mM EDTA, 1% Nonidet P-40, 1% deoxycholate) and twice with 1 ml of TE buffer. Chromatin samples were then eluted by heating for 15 min at 65°C in 300 µl of elution buffer (50 mM Tris-HCl, pH 8.0, 10 mM EDTA, 1% SDS). After centrifugation, supernatants were diluted by addition of 300 µl of TE buffer and heated overnight at 65° C to reverse cross-links. RNA and proteins were sequentially degraded by addition of 30 µg of RNase A for 30 min at 37°C, and 120 µg of proteinase K for 2–3 hr at 37°C. DNA was phenol/chloroform-extracted and ethanol-precipitated in the presence of 10 µg of tRNA as a carrier. Amplification of the indicated locus was done by quantitative PCR using Green-2-go mastermix (BioBasic Inc) on Realplex2 Mastercycler (Eppendorf).

## Cell lines and western blot

CaP2 and CaP8 murine prostatic cells were provided by Dr. Hong Wu (UCLA) and maintained as described (*Jiao et al., 2007*) and confirmed to be mycoplasma negative by PCR. Growth curves of

treated cells with indicated amount of K02288 were obtained using phase object confluence as monitored every 2 hr by IncuCyte S3 live cell imaging (Essen Bioscience). Significance between genotype was assessed by one-way ANOVA using Prism 6.0 software (GraphPad). Protein extracts were prepared after 1 hr treatment with K02288 using RIPA lysis buffer (10 mM Tris–HCl (pH 8.0), 140 mM NaCl, 1% Triton X-100, 0.1% sodium dodecyl sulphate, 0.1% deoxycholate, 1 mM EDTA) supplemented with protease and phosphatase inhibitor cocktails (Roche). Proteins were immobilized on Immobilon FL PVDF membranes (Millipore) and probed with the following antibodies diluted in Odyssey blocking buffer (LI-COR): anti-pAKT S473 (1:500, CST), anti-AKT (1:500, CST), and anti-$\alpha$-TUBULIN (DM1A) (1:2000, Sigma). Rabbit and mouse IR dye secondary antibodies (LI-COR Biosciences) were used at 1:10 000. Blots were scanned with the Odyssey imaging system (LI-COR).

# Acknowledgements

We are grateful to members of the Bouchard laboratory and Drs. Luke McCaffrey, Peter Siegel, and Yojiro Yamanaka for critical reading of the manuscript. Special thanks to Dr Meinrad Busslinger (IMP, Vienna) for providing the *Gata3bio/bio* mice. We thank the Advanced BioImaging Facility (ABIF), the Flow Cytometry platform as well as the histology core facilities of McGill University for their technical support.

# Additional information

## Funding

| Funder | Grant reference number | Author |
| --- | --- | --- |
| Canadian Institutes of Health Research | MOP-130460 | Maxime Bouchard |

The funders had no role in study design, data collection and interpretation, or the decision to submit the work for publication.

## Author contributions

Mathieu Tremblay, Conceptualization, Data curation, Formal analysis, Validation, Investigation, Visualization, Writing - original draft, Writing - review and editing; Sophie Viala, Adda-Lee Graham-Paquin, Chloe Liu, Data curation, Formal analysis, Validation, Investigation, Visualization; Maxwell ER Shafer, Data curation, Software, Formal analysis, Validation, Investigation, Visualization; Maxime Bouchard, Conceptualization, Supervision, Funding acquisition, Writing - original draft, Writing - review and editing

## Author ORCIDs

Mathieu Tremblay (iD) https://orcid.org/0000-0002-5637-3549
Maxime Bouchard (iD) https://orcid.org/0000-0002-7619-9680

## Ethics

Animal experimentation: All animal procedures were approved by McGill University Animal Care Committee (Permit#2011-5954) according to the Canadian Council on Animal Care guidelines for use of laboratory animals in biological research.

## Decision letter and Author response

Decision letter https://doi.org/10.7554/eLife.54542.sa1
Author response https://doi.org/10.7554/eLife.54542.sa2

# Additional files

## Supplementary files

- Transparent reporting form

## Data availability

Data from this study are included in the manuscript and supporting files. Source data files have been provided for Figures 1ABC-2C-3ABC-4AEFJ-2S1AC-4S1AB. Sequencing data have been deposited in GEO under accession codes GSE155289.

The following dataset was generated:

| Author(s) | Year | Dataset title | Dataset URL | Database and Identifier |
|---|---|---|---|---|
| Tremblay M, Shafer ME, Bouchard M | 2020 | Gata3 controls stem/progenitor maintenance potential in prostate organoids | https://www.ncbi.nlm.nih.gov/geo/query/acc.cgi?acc=GSE155289 | NCBI Gene Expression Omnibus, GSE155289 |

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

# Appendix 1

**Appendix 1—key resources table**

| Reagent type (species) or resource | Designation | Source or reference | Identifiers | Additional information |
|---|---|---|---|---|
| Strain, strain background (*Mus musculus*) | ARR2-PB-Cre (Pbsn-Cre), C57BL/6 (*Mus musculus*) | *Wu et al., 2001* | RRID: IMSR_JAX:026662 | (*Mus musculus*; male) |
| Strain, strain background (*Mus musculus*) | Gata3flox, C57BL/6 (*Mus musculus*) | *Grote et al., 2006* | | (*Mus musculus*; male) |
| Strain, strain background (*Mus musculus*) | Gata3GFP, C57BL/6 (*Mus musculus*) | *Grote et al., 2006* | | (*Mus musculus*; male) |
| Strain, strain background (*Mus musculus*) | Rosa26GATA3, C57BL/6 (*Mus musculus*) | *Grote et al., 2006* | | (*Mus musculus*; male) |
| Strain, strain background (*Mus musculus*) | Gata3bio, C57BL/6 (*Mus musculus*) | Dr. Busslinger, IMP, Vienna | | (*Mus musculus*; male) |
| Strain, strain background (*Mus musculus*) | Rosa26BirA, C57BL/6 (*Mus musculus*) | *Wood et al., 2016* | RRID: IMSR_JAX:010920 | (*Mus musculus*; male) |
| Strain, strain background (*Mus musculus*) | Krt5CreERT2, C57BL/6 (*Mus musculus*) | *Van Keymeulen et al., 2011* | RRID: IMSR_JAX:029155 | (*Mus musculus*; male) |
| Strain, strain background (*Mus musculus*) | Krt8CreERT2, C57BL/6 (*Mus musculus*) | *Choi et al., 2012* | | (*Mus musculus*; male) |
| Strain, strain background (*Mus musculus*) | Rosa26LstopLTdTomato, C57BL/6 (*Mus musculus*) | *Madisen et al., 2010* | RRID: IMSR_JAX:007909 | (*Mus musculus*; male) |
| Strain, strain background (*Mus musculus*) | Ptenflox , C57BL/6 (*Mus musculus*) | *Trotman et al., 2003* | | (*Mus musculus*; male) |
| Strain, strain background (*Mus musculus*) | Bmp5SE, C57BL/6 (*Mus musculus*) | *Kingsley et al., 1992* | RRID: IMSR_JAX:000056 | (*Mus musculus*; male) |
| Strain, strain background (*Mus musculus*) | SCID-beige, C57BL/6 (*Mus musculus*) | Charles River | | (*Mus musculus*; male) |
| Cell line (*Mus musculus*) | CaP2 (*Mus musculus*) | *Jiao et al., 2007* | | male prostate |
| Cell line (*Mus musculus*) | CaP8 (*Mus musculus*) | *Jiao et al., 2007* | | male prostate |
| Transfected construct (Human) | BMP5_TRC3 LentiORF puromycin V5 (pLX317) (human) | Sigma-Aldrich | TRCN0000472902 | mouse UGSM |
| Transfected construct (Human) | TRC3 LentiORF puromycin V5 (pLX317) empty (human) | Sigma-Aldrich | | mouse UGSM |
| Antibody | CD45 (30-F11; rat monoclonal) | Biolegend | | (1/500) |
| Antibody | TER119 (TER-119; rat monoclonal) | Biolegend | | (1/1,000) |

*Continued on next page*

*Appendix 1—key resources table continued*

| Reagent type (species) or resource | Designation | Source or reference | Identifiers | Additional information |
|---|---|---|---|---|
| Antibody | CD31 (MEC13; rat monoclonal) | Biolegend | | (1/1,000) |
| Antibody | CD49f (GoH3; rat monoclonal) | Biolegend | | (1/2,000) |
| Antibody | EpCAM (G8.8; rat monoclonal) | Biolegend | | (1/500) |
| Antibody | SCA1 (D7; rat monoclonal) | Biolegend | | (1/500) |
| Antibody | TROP2 (goat polyclonal) | R&D | | (1/100) |
| Antibody | KRT5 (chicken polyclonal) | Biolegend | 905901 | (1/500) |
| Antibody | KRT8/18 (guinea pig polyclonal) | Fitzgerald | 20R-CP004 | (1/200) |
| Antibody | GATA3 (H-48; rabbit polyclonal) | SantaCruz | sc-9009 | (1/100) |
| Antibody | Ki67 (SolA15, rat monoclonal) | eBioscience | 14-5698-82 | (1/200) |
| Antibody | pAKT S473 (rabbit monoclonal) | CST | 4060 | (1/500) |
| Antibody | AKT (rabbit polyclonal) | CST | 9272 | (1/500) |
| Antibody | α-TUBULIN (DM1A; mouse monoclonal) | Sigma | T9026 | (1/2000) |
| Antibody | IRDye 800CW Donkey anti-Rabbit IgG | licor | 926-32213 | (1/10,000) |
| Antibody | IRDye 680RD Donkey anti-Mouse IgG | licor | 926-68072 | (1/10,000) |
| Recombinant DNA reagent | MISSION pLKO.1-Puro shRNA scrambled (mouse) | Sigma-Aldrich | | |
| Recombinant DNA reagent | MISSION pLKO.1-Puro shRNA Bmp5 (mouse) | Sigma-Aldrich | TRCN0000065609 | |
| Recombinant DNA reagent | MISSION pLKO.1-Puro shRNA Bmp5 (mouse) | Sigma-Aldrich | TRCN0000065610 | |
| Recombinant DNA reagent | MISSION pLKO.1-Puro shRNA Bmp5 (mouse) | Sigma-Aldrich | TRCN0000065611 | |
| Sequence-based reagent | Bmp5_Fw | Sigma-Aldrich | | TTACTTAGGGGTATTGTGGGCT |
| Sequence-based reagent | Bmp5_Rv | Sigma-Aldrich | | CCGTCTCTCATGGTTCCGTAG |
| Sequence-based reagent | Gata3_Fw | Sigma-Aldrich | | GTGGTCACACTCGGATTCCT |
| Sequence-based reagent | Gata3_Rv | Sigma-Aldrich | | GCAAAAAGGAGGGTTTAGGG |
| Sequence-based reagent | Krt5_Fw | Sigma-Aldrich | | GAGATCGCCACCTACAGGAA |
| Sequence-based reagent | Krt5_Rv | Sigma-Aldrich | | TCCTCCGTAGCCAGAAGAGA |
| Sequence-based reagent | Krt14_Fw | Sigma-Aldrich | | CCTCTGGCTCTCAGTCATCC |

*Continued on next page*

*Appendix 1—key resources table continued*

| Reagent type (species) or resource | Designation | Source or reference | Identifiers | Additional information |
|---|---|---|---|---|
| Sequence-based reagent | Krt14_Rv | Sigma-Aldrich | | TGAGCAGCATGTAGCAGCTT |
| Sequence-based reagent | Krt18_Fw | Sigma-Aldrich | | ACTCCGCAAGGTGGTAGATGA |
| Sequence-based reagent | Krt18_Rv | Sigma-Aldrich | | TCCACTTCCACAGTCAATCCA |
| Sequence-based reagent | Krt8_Fw | Sigma-Aldrich | | CAAGGTGGAACTAGAGTCCCG |
| Sequence-based reagent | Krt8_Rv | Sigma-Aldrich | | CTCGTACTGGGCACGAACTTC |
| Sequence-based reagent | Trp63_Fw | Sigma-Aldrich | | AGCAGCAAGTATCGGACAGC |
| Sequence-based reagent | Trp63_Rv | Sigma-Aldrich | | CGTCTCACGACCTCTCACTG |
| Sequence-based reagent | Ly6a_Fw | Sigma-Aldrich | | CCATCAATTACCTGCCCCTA |
| Sequence-based reagent | Ly6a_Rv | Sigma-Aldrich | | GGCAGATGGGTAAGCAAAGA |
| Sequence-based reagent | Itga6_Fw | Sigma-Aldrich | | CGCTGCTGCTCAGAATATCA |
| Sequence-based reagent | Itga6_Rv | Sigma-Aldrich | | AAGAACAGCCAGGAGGATGA |
| Sequence-based reagent | Epcam_Fw | Sigma-Aldrich | | GCTGTCATTGTGGTGGTGTC |
| Sequence-based reagent | Epcam_Rv | Sigma-Aldrich | | CACGGCTAGGCATTAAGCTC |
| Sequence-based reagent | Ppia_Fw | Sigma-Aldrich | | GTGCCAGGGTGGTGACTTTACACG |
| Sequence-based reagent | Ppia_Rv | Sigma-Aldrich | | TCCCAAAGACCACATGCTTGCCA |
| Sequence-based reagent | Bactin_Fw | Sigma-Aldrich | | TTGCTGACAGGATGCAGAAGGAGA |
| Sequence-based reagent | Bactin_Rv | Sigma-Aldrich | | ACTCCTGCTTGCTGATCCACATCT |
| Sequence-based reagent | Bmp5_prom_Fw-8450 | Sigma-Aldrich | | TCGGGTGGACCAGATTTAAG |
| Sequence-based reagent | Bmp5_prom_Rv-8390 | Sigma-Aldrich | | CAGCCATTCACGAAGTTCTCT |
| Sequence-based reagent | Bmp5_prom_Fw-5942 | Sigma-Aldrich | | TGAAAGTGGAGATGGGGAAG |
| Sequence-based reagent | Bmp5_prom_Rv-5827 | Sigma-Aldrich | | CCCAGTTTTGGAGGTTCAGA |
| Sequence-based reagent | Bmp5_prom_Fw+11268 | Sigma-Aldrich | | AAAGGGAAAAGTGCTCACCA |
| Sequence-based reagent | Bmp5_prom_Rv+11312 | Sigma-Aldrich | | TCCTCCCTCAGCTCAAAGAA |
| Sequence-based reagent | Bmp5_prom_Fw+11859 | Sigma-Aldrich | | TTGGAAGAGTTCCGATGAGG |
| Sequence-based reagent | Bmp5_prom_Rv+11947 | Sigma-Aldrich | | CAGAGTGGGTGGCAACTTCT |

*Continued on next page*

*Appendix 1—key resources table continued*

| Reagent type (species) or resource | Designation | Source or reference | Identifiers | Additional information |
|---|---|---|---|---|
| Sequence-based reagent | Bmp5_prom_Fw+24561 | Sigma-Aldrich | | GTGAGGTGGCTCAGCATGTA |
| Sequence-based reagent | Bmp5_prom_Rv+24607 | Sigma-Aldrich | | CCAGGGATGGATCTCAGGT |
| Sequence-based reagent | Cyp19a1_prom-322_Fw | Sigma-Aldrich | | GCAAATGCTGCTGATGAAAT |
| Sequence-based reagent | Cyp19a1_prom-207_Rv | Sigma-Aldrich | | ACCTTATCATCTCGCCCTTG |
| Sequence-based reagent | Cdh1_prom-175_Fw | Sigma-Aldrich | | GAACGACCGTGGAATAGGAA |
| Sequence-based reagent | Cdh1_prom-98_Rv | Sigma-Aldrich | | TCCTCCACCCCTGTCTGTAG |
| Sequence-based reagent | R26wt_geno_Fw | Sigma-Aldrich | | AAGGGAGCTGCAGTGGAGTA |
| Sequence-based reagent | R26wt_geno_Rv | Sigma-Aldrich | | CCGAAAATCTGTGGGAAGTC |
| Sequence-based reagent | R26Tomato_Fw | Sigma-Aldrich | | CCCCGTAATGCAGAAGAAGA |
| Sequence-based reagent | R26Tomato_Rv | Sigma-Aldrich | | GAGGTGATGTCCAGCTTGGT |
| Sequence-based reagent | Bmp5wt_geno_Fw | Sigma-Aldrich | | TAAGGACAAGGGAAACCCTC |
| Sequence-based reagent | Bmp5SE_geno_Fw | Sigma-Aldrich | | TAAGGACAAGGGAAACCCTT |
| Sequence-based reagent | Bmp5_geno_Rv | Sigma-Aldrich | | GAACCATTTCACCAGCTCCT |
| Sequence-based reagent | Rosa26Gata3-wt_geno_Fw | Sigma-Aldrich | | AAAGTCGCTCTGAGTTGTTAT |
| Sequence-based reagent | Rosa26Gata3_geno_Rv | Sigma-Aldrich | | GCGAAGAGTTTGTCCTCAACC |
| Sequence-based reagent | Rosa26wt_geno_Rv | Sigma-Aldrich | | GGAGCGGGAGAAATGGATATG |
| Sequence-based reagent | Gata3_flox_geno_Fw | Sigma-Aldrich | | GTCAGGGCACTAAGGGTTGTT |
| Sequence-based reagent | Gata3_flox_geno_Rv | Sigma-Aldrich | | TGGTAGAGTCCGCAGGCATTG |
| Sequence-based reagent | Gata3GFP_geno_Fw | Sigma-Aldrich | | GGCCTACCCGCTTCCATTGCT |
| Sequence-based reagent | Gata3GFP_geno_Rv | Sigma-Aldrich | | TATCAGCGGTTCATCTACAGC |
| Sequence-based reagent | Pten_flox-wt_geno_Fw | Sigma-Aldrich | | AAAAGTTCCCCTGCTGATGATTTGT |
| Sequence-based reagent | Pten_flox-wt_geno_Rv | Sigma-Aldrich | | TGTTTTTGACCAATTAAAGTAGGCTG |
| Sequence-based reagent | Cre_geno_Fw | Sigma-Aldrich | | AGGTGTAGAGAAGGCACTTAGC |
| Sequence-based reagent | Cre_geno_Rv | Sigma-Aldrich | | CTAATCGCCATCTTCCAGCAGG |
| Peptide, recombinant protein | EGF | Peprotech | 315-09 | 10 ng/ml |

*Continued on next page*

*Appendix 1—key resources table continued*

| Reagent type (species) or resource | Designation | Source or reference | Identifiers | Additional information |
|---|---|---|---|---|
| Peptide, recombinant protein | hBMP5 | Aviscera Bioscience | 00013-01-100 | 50 ng/ml |
| Peptide, recombinant protein | hNOGGIN | RayBiotech | 230-00704-100 | 180 ng/ml |
| Peptide, recombinant protein | Collagenase/hyaluronidase | Stemcell Technologies | . 07912 | |
| Peptide, recombinant protein | dispase II | Roche | 4942078001 | (5 U/ml) |
| Peptide, recombinant protein | DNase I | Roche | 11284932001 | 0.1 mg/ml |
| Peptide, recombinant protein | matrigel | Corning | CACB354234 | |
| Peptide, recombinant protein | BPE | Life technologies | 13028014 | 25 ug/ml |
| Peptide, recombinant protein | b27 SUPPLEMENT | Life technologies | 17504044 | |
| Peptide, recombinant protein | collagenase A | Bioshop | COL004 | |
| Peptide, recombinant protein | fibronectin | Sigma-Aldrich | 11051407001 | |
| Peptide, recombinant protein | collagen type I | Corning | 354236 | |
| Peptide, recombinant protein | proteinase K | biobasic | PB0451 | |
| Peptide, recombinant protein | RNase A | Roche | 10109169001 | |
| Commercial assay or kit | In Situ Cell Death Detection Kit | Roche | 12156792910 | |
| Commercial assay or kit | P1 Primary Cell 4D-Nucleofector Kit | Bioscience | V4XP-1024 | |
| Commercial assay or kit | RNeasy micro kit | Qiagen | 74004 | |
| Commercial assay or kit | Green-2-go mastermix | BioBasic | QPCR004 | |
| Commercial assay or kit | TruSeq Stranded Total RNA Sample Preparation Kit | Illumina | | |
| Chemical compound, drug | tamoxifen | Toronto Research Chemicals | T006000 | 3 mg |

*Continued on next page*

*Appendix 1—key resources table continued*

| Reagent type (species) or resource | Designation | Source or reference | Identifiers | Additional information |
|---|---|---|---|---|
| Chemical compound, drug | K02288 | AdooQ Bioscience | A14311 | 10 uM |
| Chemical compound, drug | Y-27632 | ApexBio | A3008 | 10 uM |
| Chemical compound, drug | 4-hydroxy-tamoxifen | Toronto Research Chemicals | H954725 | 500 ng/ml |
| Chemical compound, drug | 5α-dihydrotestosterone | steraloids | A2570-000 | 1nM |
| Chemical compound, drug | phosphatase inhibitor cocktails (Phostop) | Sigma | 4906845001 | |
| Chemical compound, drug | Fixable Viability Dye eFluor 780 | ebioscience | 65-0865-18 | |
| Software, algorithm | Prism 6.0 | GraphPad | | |
| Software, algorithm | ImageJ | Fiji | | |
| Software, algorithm | FlowJo | LLC | | |
| Software, algorithm | DIVA | BD Biosciences | | |
| Software, algorithm | L-calc | Stem cell technologies | | |
| Software, algorithm | matrix2png program | *Pavlidis and Noble, 2003* | | |

