## [Decision Letter]

**Acceptance summary:**

Through a combination of genetic and pharmacologic approaches, your work convincingly demonstrates a role of GATA3 in regulating self-renewal of prostate basal epithelial cells, largely through transcriptional repression of BMP5. Interestingly, this GATA3/BMP5 regulatory circuit is restricted to basal cells and is not present in luminal cells. A second finding is the importance of BMP5 in the oncogenic phenotype of *Pten* loss, based on reduced levels of prostate intraepithelial neoplasia (PIN) in the setting of genetic BMP5 deletion or pharmacologic inhibition. An important remaining question for future work is whether this BMP5 dependency is a consequence of perturbation of the GATA3/BMP5 regulatory circuit (in basal cells) or through BMP5 expression in luminal cells.

**Decision letter after peer review:**

Thank you for submitting your article "Regulation of stem/progenitor cell maintenance by BMP5 in prostate homeostasis and cancer initiation" for consideration by *eLife*. Your article has been reviewed by two peer reviewers, and the evaluation has been overseen by Charles Sawyers as Reviewing Editor and Anna Akhmanova as the Senior Editor. The following individual involved in review of your submission has agreed to reveal their identity: Dean Tang.

The reviewers have discussed the reviews with one another and the Reviewing Editor has drafted this decision to help you prepare a revised submission.

Summary:

The authors provide evidence that implicates BMP5 (positively) and *Gata3* (negatively) in regulating prostate stem/progenitor cells in the context of both homeostasis and tumor development. This group has been investigating *Gata3* in both normal prostate and prostate cancer (e.g., Nguyen et al., 2013; Shafer et al., 2017). Here, the authors implicate BMP5 as a key regulator of basal prostate stem cell homeostasis. They provide data that *Bmp5* gene inactivation or chemical-mediated inhibition of Bmp signaling delays prostate and skin tumorigenesis initiated by *Pten* loss.

Essential revisions:

1) The use of Noggin to implicate BMP5 is not definitive because Noggin is not specific for *Bmp5* inhibition. We would like to see confirmatory data with *Bmp5* shRNA to corroborate the role of the *Gata3*-*Bmp5* signaling axis in organoid formation.

2) Figures 3C and 4H, the results convincingly demonstrate that decreased *Bmp5* signaling attenuated prostate regeneration and PIN progression. But these data need to be better connected to the GATA3-*Bmp5* signaling axis studies presented in the first part of the study (Figures 1 and 2). For example, do *Gata3* KO basal cells possess a higher regenerative capacity in vivo, like the basal cells stimulated with ectopic *Bmp5*?

In addition, since upregulation of *Bmp5* only happened when *Gata3* was deleted in basal cells (SF2a), the authors should comment (with additional data or in the Discussion) on whether *Gata3* KO/overexpression will impact PIN progression in the *Keratin8-CreER-Pten* (luminal Cre) model? This is an important point regarding whether the *Gata3*-*Bmp5* signaling axis is likely to play a critical role in prostate cancer progression.

---

## [Author Response]

Essential revisions:1) The use of Noggin to implicate BMP5 is not definitive because Noggin is not specific for Bmp5 inhibition. We would like to see confirmatory data with Bmp5 shRNA to corroborate the role of the Gata3-Bmp5 signaling axis in organoid formation.

In support of a specific implication of *Bmp5*, we added several new results showing that *Bmp5* inhibition or loss by either inhibitor treatment, knockout, knockdown strategy impairs the propagation potential of wild type as well as *Gata3*-deficient organoid cultures:

First, we treated our culture with a selective inhibitor of BMPR-SMAD1/5/8 signaling, K02288, which validate the effect seen by Noggin treatment (Figure 3B, Figure 3—figure supplement 2B).

Second, we created a new line of mice where both *Bmp5* and *Gata3* can be knocked-out in KRT5+ basal cells. We see that *Bmp5* mutation impaired the increased organoid-forming capacity of *Gata3*-deficient cells which clearly demonstrate that *Gata3*-deficient phenotype is specifically dependent on the BMP5 ligand (Figure 3C, Figure 3—figure supplement 2C).

Finally, we also provide evidences that the acute loss of *Bmp5* using shRNA against *Bmp5* (knockdown) showed a similar effect, inhibiting the passaging capacity of both wild type and *Gata3*-deficient organoids (Figure 3D, Figure 3—figure supplement 2D). Together, these results clearly confirm that the increased propagation potential of *Gata3*-deficient basal cells requires BMP signaling driven by the BMP5 ligand.

2) Figures 3C and 4H, the results convincingly demonstrate that decreased Bmp5 signaling attenuated prostate regeneration and PIN progression. But these data need to be better connected to the GATA3-Bmp5 signaling axis studies presented in the first part of the study (Figures 1 and 2). For example, do Gata3 KO basal cells possess a higher regenerative capacity in vivo, like the basal cells stimulated with ectopic Bmp5?

In the revised version we added several new results showing that the *Gata3*-*Bmp5* axis is link to an increase in regenerative potential and correction of prostate cancer progression.

First, as presented before in Figure 1C, *Gata3* loss in sorted basal cells using the *Pb^Cre^* driver show a higher regenerative capacity compared to wild type cells in vivo.

Second, we provide new results showing that the specific loss of *Gata3* in basal cells using *Krt5^CreERT2^* driver (PRU frequency of 1 in 1,879, Figure 1C) lead to a 4-fold increase in regenerative potential similar to basal cells stimulated with ectopic BMP5 (PRU frequency of 1 in 2,059, Figure 3E).

Third, we provide new organoid culture results showing the effect of *Bmp5* inhibition or loss by either inhibitor treatment, knockout, knockdown strategy on the propagation potential of wild type as well as *Gata3*-deficient organoid cultures (Figure 3B-D, Figure 3—figure supplement 2B-D, see comment above).

Finally, we created a new line of mice where GATA3 is overexpress together with *Pten* loss specifically in basal cells (*Krt5^CreERT2^Pten^f/f^ R26^G3/+^*). Similar to the loss of *Bmp5*, overexpression of *Gata3* corrects *Pten*-deficient tumor phenotype in both the prostate and the skin (Figure 4H-L, Figure 4—figure supplement 2D) indicating that both *Gata3* and BMP5 are key players in *Pten*-deficient cancer progression.

In addition, since upregulation of Bmp5 only happened when Gata3 was deleted in basal cells (SF2a), the authors should comment (with additional data or in the Discussion) on whether Gata3 KO/overexpression will impact PIN progression in the Keratin8-CreER-Pten (luminal Cre) model? This is an important point regarding whether the Gata3-Bmp5 signaling axis is likely to play a critical role in prostate cancer progression.

We deliberately decided to focus on basal cells based on the basal-specific response of *Bmp5* to the loss of *Gata3* (Figure 2—figure supplement 2A). By covering luminal cells, our manuscript would have lost in cohesion and depth. In support of a role for *Bmp5* in the luminal compartment, we treated *Krt8^CreERT2^Pten^f/f^* mice with the selective inhibitor of BMPR-SMAD1/5/8 signaling, K02288, for 72 days after tamoxifen induction. Our preliminary results show a correction of prostate phenotype (see Author response image 1). These results suggest strongly that GATA3 and BMP5 also play a role in luminal cells but this will have to be addressed in a separate manuscript and include a different set of experiments. We therefore added a section about this in the Discussion as suggested.

**Author response image 1. sa2fig1:** *Krt8^CreERT2^Pten^f/f^* tamoxifen-treated mice were injected with either K02288 or PBS for 10 weeks. Showed are representative histological sections of prostate tissue stained with H&E showing an absence of PIN in K02288-treated as compared to mock-treated mice.